# Design and Implementation of a High-Throughput Digital Microfluidic System Based on Optimized YOLOv8 Object Detection

**DOI:** 10.3390/mi16050521

**Published:** 2025-04-28

**Authors:** Ming Cao, Wufeng Duan, Zuwei Huang, Huihong Liang, Fanrong Ai, Xianming Liu

**Affiliations:** 1School of Advanced Manufacturing, Nanchang University, Nanchang 330031, China; caoming@ncu.edu.cn (M.C.); 402400230053@email.ncu.edu.cn (W.D.); zuweihuang@email.ncu.edu.cn (Z.H.); huihongliang@email.ncu.edu.cn (H.L.); 2Key Laboratory of Phytochemistry and Natural Medicines, Dalian Institute of Chemical Physics, Chinese Academy of Sciences, Dalian 116024, China

**Keywords:** parallel movement, GAM_Attention, EMA, YOLOv8 object detection, digital microfluidic system

## Abstract

To address the challenges of excessive control pins and inefficient high-throughput droplet manipulation in conventional digital microfluidic chips, this study developed a parallel-motion digital microfluidic system integrated with an image acquisition device. The system employs an enhanced YOLOv8 object detection model for droplet recognition. By enabling parallel droplet transportation and processing, it significantly improves operational efficiency and detection accuracy. For droplet recognition, the YOLOv8 model was optimized through the integration of GAM_Attention and EMA mechanisms, which strengthen feature extraction capabilities and detection performance. Experimental results demonstrated that the optimized model achieves remarkable accuracy and robustness in droplet detection tasks, with mAP50 increasing from 96.5% to 98.7% and mAP50–90 improving from 65.8% to 68.5%. The system exhibits enhanced detection precision and real-time responsiveness, maintaining an error rate below 0.53%. Furthermore, a host computer interface was implemented for multi-droplet path planning and feedback, establishing a closed-loop control system. This work provides an efficient and reliable solution for high-throughput operations in microfluidic chip applications.

## 1. Introduction

Digital microfluidic (DMF) systems have become a mainstream technology for droplet manipulation, owing to their unique flexibility and discrete fluid control capabilities. These systems offer a high surface-to-volume ratio (which reduces reaction times) while offering high controllability in sample handling [1]. In particular, DMF systems leveraging the Electrowetting on Dielectric (EWOD) principle utilize voltage modulation to control changes in the contact angle of droplets on hydrophobic surfaces, enabling droplet manipulation across scales ranging from nanoliters (nL) to microliters (μL).

In contrast to the fabrication of microchannels through microfluidic channels for the movement of droplets, dielectrophoresis utilizes non-uniform electric fields for control [2]. This technology exhibits advantages such as miniaturization, programmability, parallelization, and low power consumption [3]. Consequently, DMF systems provide a rational platform for the large-scale implementation of complex stepwise biological tests, making them suitable for clinical point-of-care-testing (POCT) [4], cellular biology research, environmental monitoring, and other applications.

Digital microfluidic (DMF) systems consist of various modules, including control circuit boards, DMF chips, ITO layers, and more. A DMF chip, composed of a dielectric layer, electrode layer, and substrate, typically exists in two formats: single-plate and dual-plate, also referred to as open and closed systems, respectively. The chip surface is populated with numerous electrodes, and droplet motion is achieved by applying a high voltage to designated electrodes to drive droplets to specified locations [5].

Since the 1990s, DMF systems have evolved significantly. Initially limited to controlling simple droplet movements, they are now integrated with deep learning techniques, enabling droplet recognition, tracking, and real-time feedback control. This integration enhanced the real-time controllability and accuracy of the droplet operations. For instance, Guo Kunlun et al. (2024) developed an AI-assisted multi-liquid device to improve intelligent droplet monitoring [6], while Sun Hao et al. (2023) applied deep learning for droplet control and analysis [7]. Most of these methods, however, focus on one-to-one droplet movement control. Thus, achieving high-throughput droplet manipulation while enhancing droplet recognition accuracy, speed, and real-time performance remains an urgent challenge.

In this study, we designed and developed a digital microfluidic system that supports parallel droplet movement. The system integrates a real-time monitoring camera device (as shown in Figure 1) to achieve synchronized and precise control over multiple rows of droplets, thereby significantly improving operational efficiency and response speed. In addition, the system features a host computer interface with path-planning and real-time feedback capabilities, providing users with a friendly, interactive experience and precise control.

An improved YOLOv8 model [8] was adopted for the visual detection module of the DMF system. By integrating the GAM_Attention and EMA attention mechanisms, the model’s droplet feature recognition capabilities were significantly enhanced, enabling the high-precision detection of droplet positions and real-time feedback. This improvement further enhanced the intelligence and operational performance of the system.

The entire system is equipped with efficient droplet processing, visual recognition, and extremely convenient user operation functions. It can simultaneously carry out biochemical reactions for multiple droplet groups, and users can use the upper computer to plan the movement trajectories of the droplets and check whether the trajectories are correct. It can be widely applied in medicine and other fields to achieve immediate diagnosis for patients.

## 2. Related Work

### 2.1. Mechanism of Single Droplet Actuation

Electrowetting on dielectric surfaces refers to the reduction in solid–liquid interfacial free energy between the dielectric substrate and the droplet caused by the accumulation of charges under an applied electric field. This results in a transition from a hydrophobic to a hydrophilic state on the dielectric surface. This phenomenon is known as Electrowetting on Dielectric (EWOD), or the electrowetting effect on dielectric materials [9].

The contact angle of the droplet, as well as the surface tensions among the solid, liquid, and gas phases, can be derived using Young’s equation [10,11], expressed as follows:(1)$$\gamma_{SG}-\gamma_{SL}=\gamma_{LG}\cos\theta$$

The $\gamma_{SG}$, $\gamma_{SL}$, and $\gamma_{LG}$ symbols represent the interfacial free energies of solid–gas, solid–liquid, and liquid–gas, respectively. The angle θ between $\gamma_{SL}$ and $\gamma_{LG}$ is denoted as the contact angle at the junction of the gas, liquid, and solid phases, as shown in Figure 2.

The variation in the surface tension of a liquid can be controlled by applying an external voltage. The change in dynamic surface tension can be described via the Lippmann equation. The equation is as follows:(2)$$\gamma_{SLV}=\gamma_{SL}-\varepsilon_{0}\varepsilon_{r}V^{2}/2d$$

In this equation, $\gamma_{SL}$ and $\gamma_{SLV}$ represent the surface tension between the hydrophobic solid and liquid before and after the applied voltage, respectively. $\varepsilon_{0}$ and $\varepsilon_{r}$ are the vacuum permittivity and the effective dielectric constant of the dielectric material, respectively. d is the effective thickness of the hydrophobic layer.

Due to the reduction in the surface tension of the hydrophobic solid–liquid interface, the three-phase contact angle $\theta_{V}$ of the droplet on the hydrophobic surface decreases, as shown in Figure 2. The three-phase contact angle after the applied voltage can be derived from the above two equations and is represented by the Lippmann–Young equation as follows:(3)$$\cos\theta_{V}=\cos\theta_{0}+\varepsilon_{0}\varepsilon_{r}V^{2}/2d\gamma_{LG}$$

### 2.2. Principle of Parallel Multi-Droplet Group Actuation

The EWOD system is essentially a capacitive load, and the current is primarily generated by the charging and discharging of the capacitor. The current formula for a single electrode is as follows:(4)$$I=C\frac{dV}{dt}$$

The formula for a single capacitor C is as follows:(5)$$C=\frac{\varepsilon_{0}\varepsilon_{r}A}{d}$$

In this formula, *A* represents the area of a single electrode.

If *N* electrodes are driven simultaneously, the total transient current is(6)$$I_{total}=\sum_{i=1}^{N}C_{i}\frac{dV_{i}}{dt}$$
and the total capacitance is(7)$$C_{parallel}=N\cdot C$$

The total power consumption is the average power of a single electrode multiplied by the total number of electrodes, *N*, driven in parallel, as follows:(8)$$P_{total}=\frac{1}{2}NCV^{2}f$$

In this formula, *f* is the driving frequency.

Therefore, when we know the capacitance, current, and power of a single electrode, we can calculate the total voltage required to drive the target number of parallel droplets.

### 2.3. YOLOv8 Object Detection Model and Attention Mechanism

YOLOv8 uses an improved CSPDarknet53 as the backbone network, performing five down-samplings on the input features to obtain five different-scale features, labeled as B1–B5. The original CSP (Cross Stage Partial) module in the backbone is replaced with C2f [12]. The C2f module adopts gradient parallel connections, enriching the information flow of the feature extraction network while maintaining lightweight characteristics. Inspired by PANet, YOLOv8 uses a PAN-FPN structure for the neck [13]. Compared to the necks in the YOLOv5 [14] and YOLOv7 [15] models, YOLOv8 removes the convolution operation after up-sampling in the PAN structure [13], achieving model lightweighting while maintaining the original performance. We denote the PAN and FPN structures [16] in the YOLOv8 model as P4–P5 and N4–N5, respectively, representing features at different scales. Additionally, YOLOv8 adopts a decoupled head structure for detection. The decoupled head structure uses two independent branches for object classification and bounding box regression, applying different loss functions for these two tasks. For the classification task, binary cross-entropy loss (BCE loss) is used. The formula is(9)$$BCE-Loss=-\left[y.\log(p)+(1-y).\log(1-P)\right]$$

In this equation, y is the target label. If *y* = 1, the model should output *p* as a value approaching 1; if *y* = 0, the model should output *p* as a value approaching 0. *P* is the model’s predicted value, representing the probability that the sample belongs to class 1. Additionally, 1 − *P* is the probability that the sample belongs to class 0.

For the bounding box regression task, Distribution Focal Loss (DFL) and Clou are used. The formulas are as follows:(10)$$DFL\;(p,y)=-\sum_{c=1}^{C}\left[\begin{array}{l} {\left(1-p_{c}\right)}^{\gamma }\log(p_{c})+\\ \alpha {\left(1-p_{c}\right)}^{\beta }\log(p_{c}) \end{array}\right]$$

In this equation, $p_{c}$ is the predicted probability for class c. γ and β are hyperparameters used to adjust the weight of hard and easy samples. α is a hyper parameter used to balance class imbalance.(11)$$Clou-Loss=-\sum_{c=1}^{C}\left[\alpha_{c}{\left(1-p_{c}\right)}^{\gamma }\log(p_{c})\right]$$

In this equation, $\alpha_{c}$ is the weighting coefficient for class c, typically used to adjust for class imbalance. $(1-p_{c}{)}^{\gamma }$ is the component in focal loss that reduces the impact of easily classified samples. $p_{c}$ is the model’s predicted probability for class c. γ is the focal loss hyperparameter that controls the focus on hard-to-classify samples.

In Figure 3, the structure of the YOLOv8 object detection model is illustrated [17].

In the YOLO model, to selectively focus on certain parts of the information while ignoring other visible information, an attention mechanism is typically incorporated. This improves the model’s accuracy in recognizing specific targets [18,19]. In Table 1, the four classification criteria of the attention mechanism and the attention types under each criterion are presented.

The attention mechanisms based on different criteria are not mutually exclusive; therefore, a single attention model may combine multiple criteria, as shown in Table 2.

## 3. Method

### 3.1. Design of Digital Microfluidic Chips

Typical substrate materials for digital microfluidic (DMF) chips include glass, polydimethylsiloxane (PDMS), and polymers like PMMA. PDMS, due to its transparency, ease of fabrication, and biocompatibility, has become a commonly used substrate material [20].

The fabrication techniques for microfluidic chips mainly include photolithography and etching, hot pressing, and molding methods [21]. Conventional DMF chips are generally limited to controlling the movement of a single droplet. In scenarios requiring high-throughput droplet reactions, they cannot meet the demand. This paper introduces an innovative electrode pin control method on the DMF chip, allowing multiple droplets to move in parallel. Under ideal conditions, there are no limitations on the size of the DMF chip, the number of control pins, etc., allowing for the parallel movement of an unlimited number of droplets, greatly improving the efficiency of biochemical reactions.

Specifically, the DMF chip includes both a reagent reaction and reagent embedding section. In the reagent embedding section, we can add the required reagents or droplets and then move them to the reagent reaction section for the necessary reactions. The typical design of the reagent reaction chip uses a 2 mm × 2 mm electrode as the minimum unit for droplet movement. The size of this electrode can be adjusted according to the specific application while meeting the processing requirements. Then, we arrange M single electrodes in parallel to form one row, creating a single row of M columns of electrodes. To ensure the free longitudinal movement of droplets within the channel, we must use K single rows of electrodes to form a sample movement channel, with the condition that K ≥ 3. Therefore, a sample channel is composed of a K-row and M-column electrode array. Next, if the number of samples to be processed in the actual application is N, we design N sample channels, each consisting of K rows and M columns of electrodes, ultimately forming an N × K-row × M-column electrode matrix. Finally, for each channel, the electrodes in the k-th row (k ≤ K) and m-th column (m ≤ M) are connected in parallel with the corresponding electrodes in the same row and column of the other N − 1 channels, forming a parallel electrode drive. In this way, for the reagent reaction chip with an N × K-row × M-column electrode matrix, we only need K*M control lines for driving, and the number of control lines is independent of the number of channels, ensuring scalability for different applications. In Figure 4, the electrode pin arrangement of the innovative chip and how the droplets move are illustrated.

### 3.2. Fabrication of Digital Microfluidic Chips

In the fabrication process of digital microfluidic chips, a circuit schematic design method based on a parallel pin layout is employed, and standardized PCB fabrication processes are used to achieve the mass production of the chips. The chip substrate is made of an FR-4 epoxy fiberglass composite material, whose excellent dielectric properties (dielectric constant of 4.4–4.8 @ 1 MHz), high glass transition temperature (Tg ≥ 130 °C), and mechanical stability (bending strength ≥ 400 MPa) effectively meet the comprehensive requirements of microfluidic systems for substrate materials. The electrode structure is made from a copper conductive layer (thickness 35 ± 5 μm) fabricated using photolithography, with a gold/nickel/copper composite layer formed through gold plating treatment. This structure maintains high electrical conductivity (5.96 × 10^7^ S/m) while significantly improving the electrode’s oxidation resistance (no significant corrosion after 240 h of 85 °C/85%RH environmental testing). This design method supports the dynamic optimization of electrode dimensions (accuracy ± 10 μm) by adjusting photomask parameters. Combined with the standard processes of the Jialichuang PCB manufacturing platform (line width/line spacing ≥ 4 mil, minimum hole diameter ≥ 0.3 mm), it enables rapid iteration to obtain complete functional chips that meet different driving voltage requirements (0–300 Vrms).

### 3.3. Improved YOLOv8 Object Detection Model

To improve the accuracy of the YOLOv8 model in recognizing droplet targets on digital microfluidic chips, we incorporated two attention mechanisms: GAM_Attention and EMA.

GAM_Attention is an improved attention mechanism designed to enhance the model’s ability to focus on important information by adaptively adjusting the weights of different regions in the feature map. It uses weighted input feature maps to allow the model to focus on the most useful regions or features [22]. As shown in Figure 5, the conceptual diagram of GAM is presented, and the following formula can be derived:(12)$$F_{2}=M_{C}(F_{1})⊗F_{1}$$(13)$$F_{3}=M_{s}(F_{2})⊗F_{2}$$

In the equation, M_C_ and M_S_ represent the channel attention map and spatial attention map, respectively, with their submodules shown in Figure 6; ⊗ denotes element-wise multiplication.

The channel attention submodule uses 3D arrangement to preserve cross-dimensional information. Then, a Multi-Layer Perceptron (MLP) is used to amplify the cross-dimensional channel–space dependencies. The MLP follows an encoder–decoder structure, similar to BAM, with a reduction ratio of r. In the spatial attention submodule, two convolutional layers are used to fuse spatial information, and the same reduction ratio r, as in BAM, is applied to reduce the spatial information [23].

The Exponential Moving Average (EMA) attention mechanism is an improved approach that introduces the EMA method based on the traditional attention mechanism. It aims to enhance the model’s ability to handle long sequences or long-term dependencies, improving both stability and performance. It works by applying a weighted average to historical data to smooth out noise or fluctuations. In the attention mechanism, EMA can be used to smooth the updating process of attention weights. Compared to the traditional attention mechanism, which directly calculates the relationship between the current input and query to generate weights, the EMA attention mechanism applies an exponential weighted average to the historical attention weights during each update. This makes each computation more stable and less susceptible to single-point fluctuations. The schematic diagram of the EMA attention mechanism is shown in Figure 7 [24].

By synergizing the strengths of the GAM_Attention and EMA attention mechanisms, we insert them into the original YOLOv8 object detection model. The GAM_Attention mechanism is inserted before the Spatial Pyramid Pooling Fast (SPPF) layer, and the EMA attention mechanism is inserted after it. This approach helps the model selectively focus on more important feature areas, mitigating interference from extraneous background information, while also stabilizing the performance of pooled features, making the model more efficient in extracting the necessary features.

## 4. Experiment and Discussion

### 4.1. Measurement of Droplet Contact Angle

The movement of droplets on digital microfluidic chips requires treating the surface, optimizing droplet operation performance, enhancing chip reliability, and improving experimental reproducibility. Common surface treatments typically involve coating the digital microfluidic chip with a hydrophobic material, such as silicone oil, peanut oil, or Teflon, followed by applying an insulating layer material, such as PP film or ETFE film. Finally, a layer of hydrophobic material is applied on top of the insulating layer [25,26].

To explore the optimal conditions for droplet movement on digital microfluidic chips, various combinations of hydrophobic and insulating layer materials were tested for surface treatment. We then verified the contact angles of droplets of different sizes on the treated digital microfluidic chip surface at 0 V.

In Figure 8, the results of the treatments are presented. The thicknesses of the ETFE film, PP film, PE film, and PDFE film are 15 µm, 10 µm, 20 µm, and 30 µm, respectively, with 20 µL of silicone oil, peanut oil, and Soft99 applied per coating. The droplet sizes are 5 µL, 8 µL, and 10 µL. From this Figure, it can be observed that when the droplet size is 5 µL, the insulating layer material is 2 µm, and the hydrophobic layer is silicone oil, the contact angle of the droplet at 0 V on the digital microfluidic chip surface is the largest. Therefore, the improved YOLOv8 model will be used to recognize droplets under these conditions.

### 4.2. Training of the Improved YOLOv8 Object Detection Model

The original YOLOv8 object detection model is enhanced by incorporating two attention mechanisms: GAM_Attention and EMA. Real-time data collection of droplets under optimal movement conditions is performed using a Basler industrial camera, which is pre-placed on the digital microfluidic chip. The Basler industrial camera is equipped with dedicated driver software, and the real-time video signal is configured with a resolution of 1280 × 720 for color video.

In the early stages of training, we used the imaging functionality of the Basler industrial camera (Ahrensburg, Germany) to capture 1000 images of droplets distributed across different small electrode positions. The collected data were then divided into four parts for the training set and one part for the prediction set. The images in the training set were labeled, with the target objects to be identified extracted. In the captured images, a rectangular grid was drawn around the electrodes of the digital microfluidic chip, and labels were assigned to the droplets and the four corners of the rectangular grid. The improved YOLOv8 object detection model was then trained using this labeled training set. We use the Dell G15 produced in the United States. The GPU used for training was an RTX 3060, with an AMD Ryzen7 6800 CPU and 16 GB of RAM. The experiments were conducted on a Windows system using the PyTorch framework and Python version 3.10, with the model trained for 3000 epochs, taking 6 h to complete. To better highlight the superiority of the improvements made to the model, we trained the same training set under identical conditions using the original YOLOv8 object detection model, with the results shown in Figure 9. As seen in this Figure, the improved YOLOv8 model outperformed the original model, with a three-point reduction in loss, and an increase of two points in both precision and recall. Additionally, both mAP50 and mAP50–90 showed improvements, with mAP50–90 increasing by three points. The object detection algorithm achieved a processing speed of 20 frames per second [27].

Moreover, as long as the real-time captured field of view was sufficiently wide, the algorithm could theoretically recognize all the processed droplets. Additionally, the frame rate of the algorithm’s recognition did not decrease with an increase in the number of droplets.

### 4.3. Design of the Droplet Operation Interface

Using the designed digital microfluidic chip with multiple sets of droplets moving in parallel equipped with a Basler industrial camera for real-time image acquisition, high-throughput droplet operations and real-time accurate monitoring can be achieved. Additionally, to facilitate the control and feedback of droplet movement trajectories, a host software interface was designed to perform complex functions for multi-droplet operations. This software allows users to input specific rows, columns, and channel numbers, which are then displayed on the chip view. For example, when inputting a chip size of three rows, ten columns, and three channels, a 9 × 10 electrode array will appear, and the designated rows and columns for the channel number will be highlighted. When clicking on a specific single electrode, all parallel electrodes corresponding to it will be highlighted in red. Users can then set the voltage hold time and record the path for future droplet movement trajectory planning, as shown in Figure 10.

To demonstrate whether the planned multiple parallel droplet movement trajectories are feasible, trajectory demonstration animations were also added to the host interface. The software is also equipped with camera design features, allowing users to design the row and column numbers as well as the channel count based on the size of the fabricated digital microfluidic chip, ensuring that the chip view displayed is consistent with the actual design. After receiving feedback on the droplet position from the improved YOLOv8 object detection model, the corresponding electrodes in the chip view will be highlighted in red, providing a more intuitive visualization of the droplet’s actual movement to the respective electrodes.

## 5. Real-Time Detection Testing on the Upper Computer

During the testing of droplet recognition accuracy on the upper computer equipped with the improved YOLOv8 object detection model, the upper computer was directly connected to the lower computer via a Nano microcontroller, enabling real-time detection and feedback on the droplet electrode positions on the digital microfluidic chip.

As shown in Figure 11, the constructed hardware system consists of a Basler industrial camera, a digital microfluidic system, and a placement platform from top to bottom. To measure the droplet contact angle, an additional Hikvision industrial camera was configured alongside the entire hardware setup. Figure 11 clearly demonstrates that the designed enclosed system can accurately and in real time provide feedback on the droplet’s position on the electrodes.

## 6. Conclusions

In summary, this paper presents an innovative parallel digital microfluidic chip design that achieves the synchronous movement of multiple droplet groups through electrode parallel connection, providing an efficient solution for high-throughput droplet-based biochemical reactions. Using this method, the system can process up to 104 sets of droplets per minute under the single-chip limit. Compared with traditional DMF control methods, this design not only optimizes the number of control electrode pins but also simplifies circuit layout and device integration.

To further enhance the recognition accuracy of droplet positions during movement, we improved the original YOLOv8 object detection model by incorporating a GAM_Attention module and EMA attention mechanisms. These enhancements significantly strengthened the model’s perception capability for droplet localization and adaptability in dynamic environments. Experimental results demonstrate outstanding performance across multiple key metrics: the loss rate decreased by three percentage points, precision and recall rates each improved by two percentage points, and the mAP value showed notable gains, particularly with a two-percentage-point increase in the comprehensive mAP50–90 evaluation metric.

Through an intuitive host computer interface design and efficient interactive modes, users can observe droplet trajectories in real time, adjust experimental parameters promptly, and flexibly plan droplet movement paths. Simultaneously, droplet positions are fed back in real time via the model to ensure the precise execution of motion control.

Collectively, the proposed parallel digital microfluidic chip and its intelligent control–detection integrated system break through the efficiency bottlenecks of traditional droplet manipulation technologies, demonstrating revolutionary application potential in precision medicine, high-throughput drug screening, and environmental trace detection.

## Figures and Tables

**Figure 1 micromachines-16-00521-f001:**
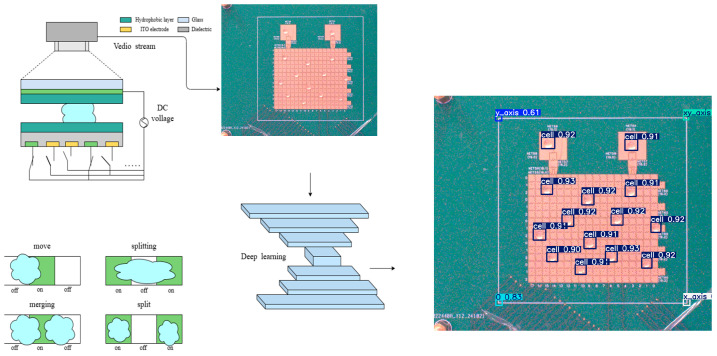
Schematic diagram of real-time droplet monitoring.

**Figure 2 micromachines-16-00521-f002:**
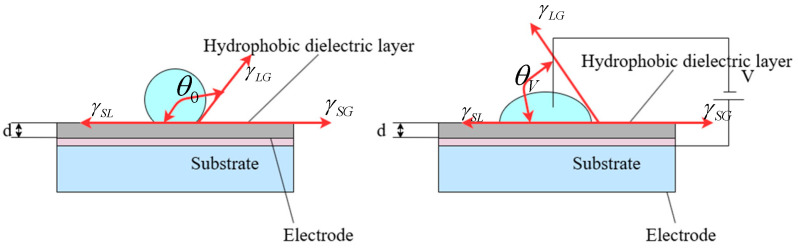
Schematic diagram of the electrowetting system on a dielectric surface.

**Figure 3 micromachines-16-00521-f003:**
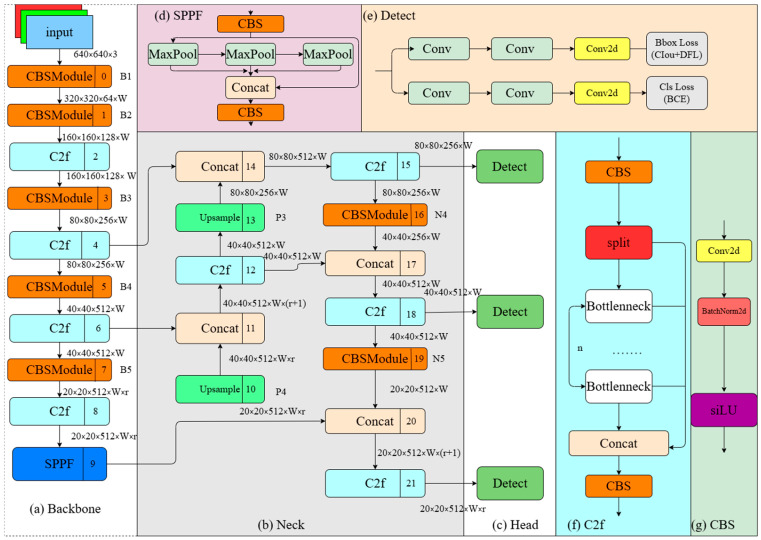
Structure diagram of the YOLOv8 object detection model.

**Figure 4 micromachines-16-00521-f004:**
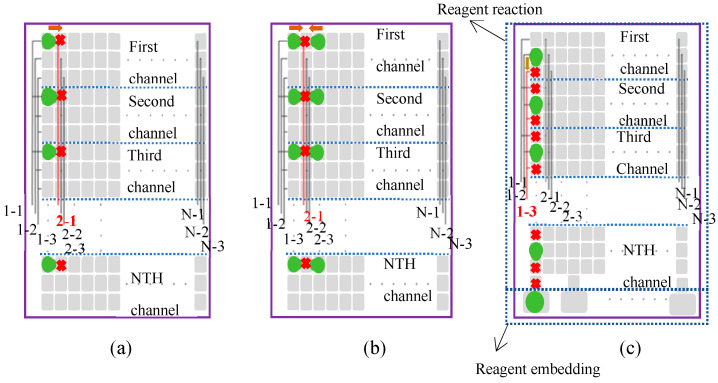
Schematic diagram of droplet movement. The green represents the droplets, and the arrows indicate the direction of droplet movement. The number–number format represents the column–row, with the first number indicating the column and the second number indicating the row, e.g., 1–2 means the first column and the second row. The wiring arrangement on each small electrode is directly represented by black lines. When the black lines turn red, it indicates that the corresponding small electrode is under high voltage, with a red cross symbolizing the electrode at high voltage. (**a**) Droplet movement to the right, with similar principles for left, up, and down movements; (**b**) droplet merging, with similar principles for droplet splitting; (**c**) droplet moving from the reagent embedding area to the reagent reaction area, with similar principles for collecting droplets from the reaction area back to the reagent embedding area.

**Figure 5 micromachines-16-00521-f005:**
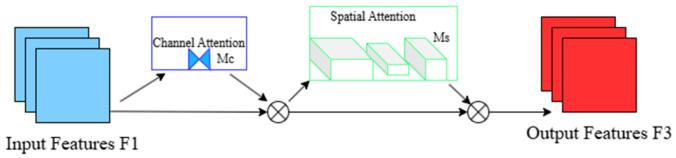
Conceptual diagram of GAM_Attention.

**Figure 6 micromachines-16-00521-f006:**
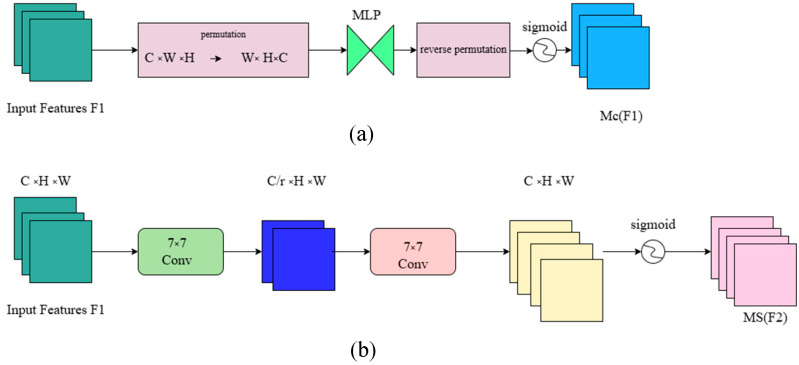
Channel and spatial submodule diagram. (**a**) represents the channel submodule diagram, and (**b**) represents the spatial submodule diagram without group convolution.

**Figure 7 micromachines-16-00521-f007:**
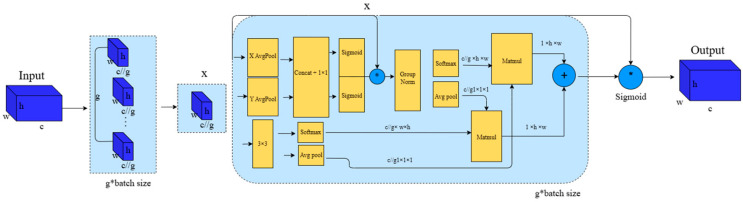
Schematic diagram of the EMA attention mechanism. Here, “g” represents the divided groups, “X Avg Pool” represents the 1D horizontal global pooling, and “Y Avg Pool” represents the 1D vertical global pooling.

**Figure 8 micromachines-16-00521-f008:**
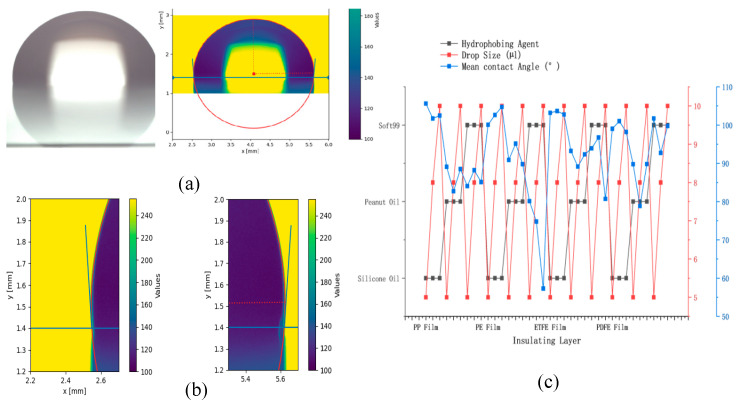
Contact angle of droplets under 0 V voltage with different surface treatments. (**a**) shows the physical image taken by a Hikvision industrial camera, along with the contact angle automatically calculated by the software program. (**b**) shows a close-up of the contact angle calculation by the software program. (**c**) displays the contact angles of droplets under different surface treatments at 0 V, where the *x*-axis represents the insulating layer material, and the three *y*-axes, from left to right, represent the hydrophobic material, droplet size, and average contact angle, respectively.

**Figure 9 micromachines-16-00521-f009:**
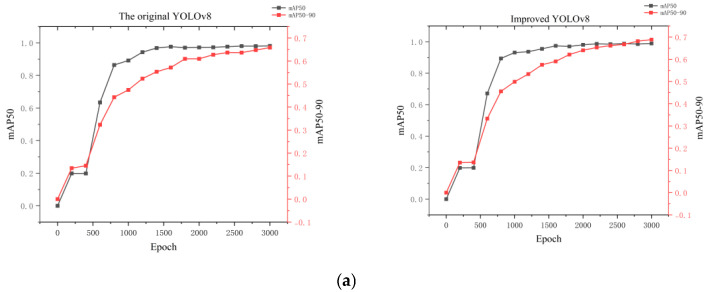
Training and evaluation of the object detection model. (**a**–**c**) represent the comparisons between the original YOLOv8 object detection model and the improved YOLOv8 model after training, showing mAP50 and mAP50–90, various losses, and accuracy and recall.

**Figure 10 micromachines-16-00521-f010:**
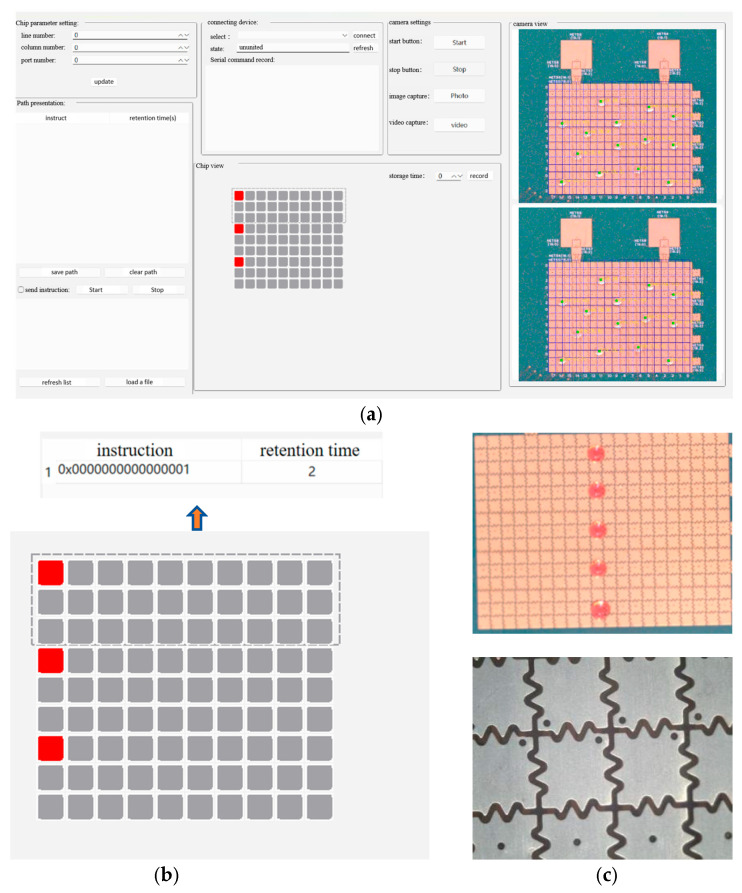
User interface of the upper computer. (**a**) represents the entire upper computer operation interface; (**b**) represents the chip electrode setup and path recording; and (**c**) represents the chip electrodes captured by the Basler industrial camera and the zoomed-in view.

**Figure 11 micromachines-16-00521-f011:**
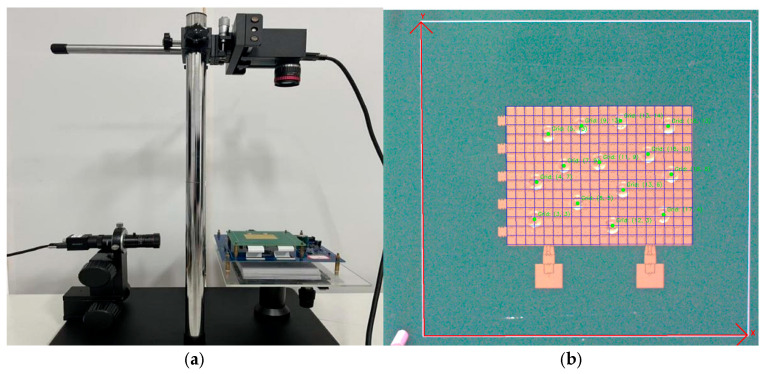
Hardware arrangement of the digital microfluidic system and droplet recognition on the upper computer. (**a**) represents the hardware setup diagram; (**b**) represents the electrode diagram of the digital microfluidic system with droplet feedback through the host computer’s vision system.

**Table 1 micromachines-16-00521-t001:** Four classification criteria of attention mechanisms and the attention types under each criterion.

Criterion	Type
Softness of Attention	Soft/hard, global/local
Forms of Input Feature	Item-wise, location-wise
Input Representations	Distinctive, self, co-attention, hierarchical
Output Representations	Single-output, multi-head, multi-dimensional

**Table 2 micromachines-16-00521-t002:** Examples of combinations between different categories.

Application	Category			
	Softness of Attention	Forms of Input Feature	Input Representations	Output Representations
Machine Translation	Soft	Item-wise	Distinctive	Single-output
Image Classification	Hard	Location-wise	Distinctive	Single-output
Machine Translation	Soft	Item-wise	Distinctive	Multi-head
Visual Question Answering	Soft	Item-wise	Co-attention and Hierarchical	Single-output
Language Understanding	Soft	Item-wise	Distinctive	Multi-dimensional
Image Classification	Soft	Location-wise	Distinctive	Single-output
Document Classification	Soft	Item-wise	Hierarchical	Single-output

## Data Availability

All the data presented in this study are available in this article.

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
