# Peer review of "Design and Implementation of a High-Throughput Digital Microfluidic System Based on Optimized YOLOv8 Object Detection"

_micromachines, 2025, doi:10.3390/mi16050521_

Round 1
Reviewer 1 Report
Comments and Suggestions for Authors
The paper presents a digital microfluidic platform incorporating YOLOv8 algorithm based object detection. The paper claims to get exceptional experimental results using the algorithm. The work is interesting and should be of interest to the microfluidic community. My main concern with the work is that the results are not well presented. No supplementary experimental video is included with the manuscript. That is essential to see the real-time detection of the droplets. Also, the figures and text do not explain well if each individual droplet is tracked for a significant duration using the proposed algorithm. Significant revision of the results section is necessary to highlight the achievements of the work more. I recommend major revisions before the paper can be considered for publication. A few of my other comments are listed below:
1. The term high-throughput is mentioned throughout the manuscript. However, no quantitative measure of throughput is mentioned in the results section. Usually, high-throughput systems refer to flowing microfluidic system were hundreds to thousands of droplets can be processed per minute. It is difficult for an electrode matrix based electrowetting system to match such throughput. In any regards, please mention how many droplets per minute can be processed by the device.
2. Section 3.1 is titled "fabrication of the digital microfluidic chips". However, it contains no information about the device fabrication. Instead, it discusses the device design. The name of this subsection should be changed. Also, a separate section about fabrication process should be included. How was the device made? What are the substrates? What metal is used for the electrodes? Is it fabricated in by the authors or bought? Is lithography used? A lot of details are missing.
3. One of the key performance terms of real-time object detection from video signal is the frame rate achieved. This information is not mentioned in the manuscript. Please mention how many frames per second can the object detection algorithm process. Also, what is the resolution of the video signal? Is it a color or black and white video?
4. Regarding the training of the algorithm, please mention the size of the training set. Also, discuss in detail the data type, size and labeling information. How long was the algorithm trained for?
5. It is mentioned that RTX 3060 GPU was used for training. However, was the same hardware used for running the algorithm? Also, other details about the computer besides the GPI information should be mentioned. For example, RAM, CPU etc. information should be included.
6. Fig. 4 caption states that the green represents the droplets. However, the figure does not contain any green objects.
7. Fig. 1, especially the DMF chip schematic, needs to be improved. The text "cell" is upside down. The resolution is also low. It is not clear what do the rectangular structures labeled "deep learning" represent. Overall, the clarity is poor. Please revise.
8. Please include experimental videos showing the droplet manipulation operation of the device as part of supplementary information.
9. What is the maximum number of droplets that the algorithm can track simultaneously? Is the frame rate reduced with the increase in number of droplets?
10. YOLO is a well-known algorithm. It is not clear what is the novelty of the work. For example, the work https://doi.org/10.1063/5.0097597 appears t discuss a similar topic. Please clearly discuss this in the introduction section.
11. Since the paper discusses digital microfluidics and droplet transport some other mechanism besides electrowetting should be briefly discussed in the introduction section. For example, dielectrophoresis techniques employing electrode array can also be used to transport droplets, particles and cells. Some references on the topic should be cited with proper discussion:
a. https://doi.org/10.1002/elps.200600549
b. https://doi.org/10.1039/B909158K
c. https://doi.org/10.1021/acs.langmuir.2c02235
Author Response
Comments 1:The term high-throughput is mentioned throughout the manuscript. However, no quantitative measure of throughput is mentioned in the results section. Usually, high-throughput systems refer to flowing microfluidic system were hundreds to thousands of droplets can be processed per minute. It is difficult for an electrode matrix based electrowetting system to match such throughput. In any regards, please mention how many droplets per minute can be processed by the device.
Response 1:In this chapter of the conclusion, we have supplemented that in the case of single-chip operation, up to 10,000 droplet processing groups can be achieved per minute.
Comments 2:Section 3.1 is titled "fabrication of the digital microfluidic chips". However, it contains no information about the device fabrication. Instead, it discusses the device design. The name of this subsection should be changed. Also, a separate section about fabrication process should be included. How was the device made? What are the substrates? What metal is used for the electrodes? Is it fabricated in by the authors or bought? Is lithography used? A lot of details are missing.
Response 2:In the paper, Section 3.2 is added to describe the fabrication process of the digital microfluidic chip in detail. It elaborates that the chip was manufactured by drawing the schematic diagram ourselves and then fabricating the board. Besides, it also explains the substrate and electrode materials used in the chip.
Comments 3:One of the key performance terms of real-time object detection from video signal is the frame rate achieved. This information is not mentioned in the manuscript. Please mention how many frames per second can the object detection algorithm process. Also, what is the resolution of the video signal? Is it a color or black and white video?
Response 3: In the section of "Training of the Improved YOLOv8 Object Detection Model", we have provided additional information that the equipment used for real-time video acquisition in this video is a BASLER industrial camera. Moreover, by using a dedicated driver software, we set the resolution of the acquired video to 1280x720 in color. Furthermore, we have also added that this object detection algorithm can process 20 frames per second.
Comments 4:Regarding the training of the algorithm, please mention the size of the training set. Also, discuss in detail the data type, size and labeling information. How long was the algorithm trained for?
Response 4: In the section "Training of the Improved YOLOv8 Object Detection Model", we have provided additional clarification that the data collected consists of 1000 images of droplets at different electrode positions. These images were equally divided into 4 training sets and 1 prediction set. The label types for the training set are 5 categories, and the training process lasted for 3000 rounds over a period of 6 hours.
Comments 5: It is mentioned that RTX 3060 GPU was used for training. However, was the same hardware used for running the algorithm? Also, other details about the computer besides the GPI information should be mentioned. For example, RAM, CPU etc. information should be included.
Response 5: In the section "Training of the Improved YOLOv8 Object Detection Model", we have supplemented that the CPU used is AMD Ryzen7 6800 and the RAM is 16G.
Comments 6: Fig. 4 caption states that the green represents the droplets. However, the figure does not contain any green objects.
Response 6: In the original draft of our paper, Figure 4 shows green color. However, we are puzzled as to why the green icon turned yellow after uploading it to our journal. We are not sure what caused this change.
Comments 7: Fig. 1, especially the DMF chip schematic, needs to be improved. The text "cell" is upside down. The resolution is also low. It is not clear what do the rectangular structures labeled "deep learning" represent. Overall, the clarity is poor. Please revise.
Response 7:We have redrawn Figure 1 and reversed the text "cell" from inverted to upright. The rectangular structure of deep learning serves as the bounding box for identifying objects. That is, in real-time recognition, if a target object is identified, a rectangular box will be selected to enclose it and the name of the target will be displayed.
Comments 8:Please include experimental videos showing the droplet manipulation operation of the device as part of supplementary information.
Response 8: We will upload the video of droplet operation in the attachment. It covers a complete set of droplet movement procedures, namely liquid injection, parallel movement, parallel merging and parallel splitting.
Comment 9:What is the maximum number of droplets that the algorithm can track simultaneously? Is the frame rate reduced with the increase in number of droplets?
Response 9:In the section of "Training of the Improved YOLOv8 Object Detection Model", we have provided additional explanations regarding the maximum number of droplets that the algorithm can track simultaneously and the variation of its frame rate with the number of droplets. Specifically, as long as the real-time collected field of view is wide enough, theoretically all the droplets processed can be identified. Moreover, the frame rate identified by the algorithm will not decrease as the number of droplets increases.
Comment 10:YOLO is a well-known algorithm. It is not clear what is the novelty of the work. For example, the work https://doi.org/10.1063/5.0097597 appears t discuss a similar topic. Please clearly discuss this in the introduction section.
Response 10: In the introduction, we elaborated on the improvements made to the traditional YOLOv8 model. By integrating GAM_Attention and EMA attention mechanisms, the recognition accuracy of droplets has been enhanced. This enables more accurate identification of droplets, thereby significantly improving the accuracy of the subsequent path planning for the upper computer's droplet operation. Moreover, it can detect droplets that deviate from the prescribed trajectory to a high degree and intervene accordingly.
Comment 11: Since the paper discusses digital microfluidics and droplet transport some other mechanism besides electrowetting should be briefly discussed in the introduction section. For example, dielectrophoresis techniques employing electrode array can also be used to transport droplets, particles and cells. Some references on the topic should be cited with proper discussion:
a. https://doi.org/10.1002/elps.200600549
b. https://doi.org/10.1039/B909158K
c. https://doi.org/10.1021/acs.langmuir.2c02235
Response 11: We have added other mechanisms in addition to electrohydrodynamics in the introduction. Since the references a and b provided are quite old at present, we only cited reference c given in the paper.

Reviewer 2 Report
Comments and Suggestions for Authors
Ming Cao et.al. developed a parallel-motion digital microfluidic system integrated with an image acquisition device, which employs an enhanced YOLOv8 object detection model for droplet recognition. This work can improves operational efficiency and detection accuracy in the parallel droplet transportation and processing. This study is interesting and meaningful. I think this work is publishable after minor revision.
(1)This work should demonstate the mian advantages and potential applications of proposed system in the introduction.
(2)Fabrication processes of microfluidic device should be given in detail.
(3)The figures should be revised, espectially the postion of ‘a’’b’’c’.
(4)I think reference should not be cited in the conclusion.
(3)Scale bar should be added in the some figures.
Comments on the Quality of English LanguageThe English could be improved to more clearly.
Author Response
Comments1:This work should demonstate the mian advantages and potential applications of proposed system in the introduction.
Response 1:In the introduction, we elaborated on the main advantages of this system and the applicable fields it can be applied to. Besides, in the conclusion section, we also emphasized these points.
Comments 2:Fabrication processes of microfluidic device should be given in detail.
Response 2:We have added Section 3.2 in the paper to provide more details on the manufacturing process and materials of the digital microfluidic chips.
Comments 3:The figures should be revised, espectially the postion of ‘a’’b’’c’.
Response 3:We have made some adjustments to the figures and repositioned a, b, and c.
Comments 4:I think reference should not be cited in the conclusion.
Response 4:We have removed the references cited in the conclusion section of the paper, so there are no references cited in the conclusion itself.
Comments 5:Scale bar should be added in the some figures.
Response 5:For the figures where adding a scale bar is appropriate, we will incorporate it.
Round 2
Reviewer 1 Report
Comments and Suggestions for Authors
The paper has been improved and can be considered for publication. I noticed some typos. I recommend a thorough proof-reading during the typesetting process.